# Recombinant Chimeric Virus-like Particles of Human Papillomavirus Produced by Distinct Cell Lineages: Potential as Prophylactic Nanovaccine and Therapeutic Drug Nanocarriers

**DOI:** 10.3390/v17091209

**Published:** 2025-09-04

**Authors:** Cyntia Silva Oliveira, Dirce Sakauchi, Érica Akemi Kavati Sasaki, Aurora Marques Cianciarullo

**Affiliations:** 1Butantan Institute, 1500 Vital Brazil Avenue, São Paulo 05503-900, SP, Brazil; dirce.sakauchi@butantan.gov.br (D.S.); eakavati@yahoo.com.br (É.A.K.S.); 2Federal University of ABC (UFABC), Santo Andre 09210-580, SP, Brazil

**Keywords:** HPV16 L1/L2 VLPs, high-risk HPV, cervical cancer, in silico epitopes evaluation, chimeric nanovaccines, nanocarriers

## Abstract

Antigenicity and immunogenicity define a potent immunogen in vaccinology. Nowadays, there are simplified platforms to produce nanocarriers for small-peptide antigen delivery, derived from various infectious agents for the treatment of a variety of diseases, based on virus-like particles (VLPs). They have good cell-penetrating properties and protective action for target molecules from degradation. Human papillomavirus (HPV) causes anogenital warts and six types of cancer in infected women, men, or children, posing a challenge to global public health. The HPV capsid is composed of viral type-specific L1 and evolutionarily conserved L2 proteins. Produced in heterologous systems, the L1 protein can self-assemble into VLPs, nanoparticles sized around 50–60 nm, used as prophylactic vaccines. Devoid of the viral genome, they are safe for users, offering no risk of infection because VLPs do not replicate. The immune response induced by HPV VLPs is promoted by conformational viral epitopes, generating effective T- and B-cell responses. Produced in different cell systems, HPV16 L1 VLPs can be obtained on a large scale for use in mass immunization programs, which are well established nowadays. The expression of heterologous proteins was evaluated at various transfection times by transfecting cells with vectors encoding codon-optimized *HPV16L1* and *HPV16L2* genes. Immunological response induced by chimeric HPV16 L1/L2 VLP was evaluated through preclinical assays by antibody production, suggesting the potential of broad-spectrum protection against HPV as a prophylactic nanovaccine. These platforms can also offer promising therapeutic strategies, covering the various possibilities for complementary studies to develop potential preventive and therapeutic vaccines with broad-spectrum protection, using in silico new epitope selection and innovative nanotechnologies to obtain more effective immunobiologicals in combating HPV-associated cancers, influenza, hepatitis B and C, tuberculosis, human immunodeficiency virus (HIV), and many other illnesses.

## 1. Introduction

Virus-like particles (VLPs) are a promising biotechnology used in vaccine and drug delivery research, as they are safe, highly immunogenic, and adaptable. VLPs play a key role in addressing HPV-related diseases, both in preventing new infections and as nanocarriers for treating existing tumor cells [1].

VLPs are impressive tools that can be applied to work with viral structures at the atomic level and have become key tools in studies in several areas, such as biology, medicine, and engineering. Thus, the use of VLPs in the development of antiviral vaccines was a natural path to follow. The success of this application paved the way for the development of other vaccine types and delivery system technologies, utilizing them as nanomachines capable of delivering drug products directly to their specific targets [2]. The application capabilities of VLPs show potential, leading to studies and investments in the development of prophylactic and therapeutic vaccines as a new generation of immunobiologicals with higher efficacy and the possibility of significant financial returns [2,3].

VLPs are multimeric nanostructures composed of viral structural proteins with self-assembling ability. They can be produced through the expression of one or more recombinant viral proteins and measure 0.1–200 nm in diameter [3]. VLPs can be defined into two classes, according to the presence or absence of a viral envelope. Enveloped VLPs (eVLPs) present a protein matrix surrounded by a residual host lipid layer, which may contain interspersed glycoproteins, such as those found in VLPs from the viruses influenza A, chikungunya (CHIKV), dengue (DENV), and others [4,5]. In turn, non-enveloped VLPs are structurally simple, consisting only of a capsid made up of viral structural proteins, as seen in HPV VLPs [5].

The eVLPs are nanostructures that mimic native enveloped virions through self-assembly of viral structural proteins but lack viral genetic material. Their morphology—typically icosahedral, spherical, or rod-shaped—is determined by the parental virus’s capsid proteins, with structural complexity ranging from single-protein assemblies to multi-layered structures incorporating matrix proteins and host-derived lipid membranes containing viral glycoproteins [6]. Unlike non-enveloped VLPs that consist solely of protein capsids and demonstrate greater stability, these lipid-enclosed particles are susceptible to environmental conditions during production and purification, where factors like temperature, shear forces, and processing methods can compromise their integrity and immunogenicity. The proper folding and glycosylation of surface antigens is critical for vaccine efficacy, as it affects antigen stability and immune recognition [7]. Production challenges include the need for specific expression systems capable of generating both internal protein cores and properly oriented envelope glycoproteins, with additional complications arising from variable protein expression levels across different systems. Large-scale manufacturing faces significant hurdles in maintaining particle purity, as host cell contaminants (including proteins, DNA, and lipids) must be removed through often costly and technically demanding purification processes [6]. These biophysical challenges are compounded in systems like baculovirus expression, where particle characteristics may lead to undesirable side effects or altered antigenicity. Despite these obstacles, strategic approaches such as introducing stabilizing mutations can improve eVLP thermostability, while optimized purification protocols help address contamination issues that could otherwise compromise vaccine safety and effectiveness [7].

The characteristics of VLPs, including their nanoscale size, multimeric antigen presentation possibilities, and highly organized repetitive structure, underscore the potential of biotechnology in vaccine generation [8]. The absence of viral genomic material is critical, ensuring they remain safe as they avoid viral infection or replication. The immune response induced by VLPs is promoted by conformational viral epitopes, which generate an effective T- and B-cell reaction (Figure 1). VLPs elicit potent humoral and cellular immune responses through their unique structural and immunological properties. Upon administration, their particulate nature and repetitive surface geometry facilitate efficient uptake by antigen-presenting cells (APCs) through phagocytosis and enhanced opsonization. At the same time, their multivalent epitope display promotes strong B cell receptor cross-linking (BCR) [9]. Within APCs, VLPs undergo proteolytic processing and achieve dual major histocompatibility complex (MHC) presentation—with MHC-II presentation activating CD4+ T cells (T helper—Th) and cross-presentation via MHC-I stimulating CD8+ T cells. This coordinated antigen presentation triggers clonal expansion of cytotoxic T lymphocytes (CTLs) capable of direct tumor cell killing, supported by Th1-derived cytokines (interleukin 2, interferon-gamma), while simultaneously driving germinal center formation through Th-B cell interactions mediated by CD40L-CD40 costimulation and cytokine signaling [10]. Robust B cell activation, facilitated both by direct VLP-BCR engagement and by Th, leads to the generation of high-affinity neutralizing antibodies from plasma cells and durable memory B cell populations. Concurrently, the immune response generates memory CTLs for long-term cellular immunity [6]. This dual activation mechanism—combining the inherent immunogenicity of particulate antigens with efficient cross-presentation capabilities—enables VLPs to stimulate comprehensive protective immunity encompassing both antibody-mediated neutralization and cytotoxic T cell responses, making them exceptionally versatile vaccine platforms.

To date, several VLP-based vaccines have received approval for both human and veterinary use, with many other candidates currently in advanced stages of clinical evaluation [12]. Moreover, the versatility of VLPs as platforms for displaying foreign antigens has significantly broadened their potential applications. This adaptability allows VLPs to function not only as prophylactic vaccines but also as drug carriers and potentially as therapeutic vaccines, paving the way for the development of next-generation vaccine technologies [13,14]. They can be produced in various systems through genetic engineering, using established cell lines from vertebrates, insects, plants, yeast, and bacteria, allowing for large-scale production for mass population immunization. For human and veterinarian use, prophylactically and/or therapeutically, VLPs provide high immunogenicity as nanovaccines and efficiency as drug nanocarriers [6,7,15].

HPV belongs to Alphapapillomavirus (α-HPV), the primary genus within the Papillomaviridae family implicated in human infections. This genus comprises approximately 65 viral types, which are categorized into low-risk HPV (LR-HPV) and high-risk HPV (HR-HPV) based on their carcinogenic potential [1]. LR-HPV encompasses 13 HPV genotypes (types 6, 6a, 6b, 11, 40, 42, 43, 44, 54, 61, 70, 72, and 81) according to epidemiological evidence [16], with HPV 6 and 11 being associated with benign conditions such as condyloma acuminatum and conjunctival papilloma [17]. Conversely, HR-HPV includes 19 HPV genotypes (types 16, 18, 26, 31, 33, 35, 39, 45, 51, 52, 53, 56, 58, 59, 66, 68a, 73, 82, and 82 subtype) [16], with HPV 16 and 18 being particularly implicated in HPV-related malignancies, such as cervical cancer [17].

HPV is a small virus characterized by an icosahedral structure with a diameter of approximately 55 nm. Its genetic composition consists of a single, circular double-stranded DNA molecule, which encodes six early regulatory proteins (E) and two late structural proteins (L). While the development of therapeutic vaccines for existing lower genital tract HPV infections remains a significant challenge, remarkable progress has been made in prophylactic vaccination. The L1 structural protein is well-known for its ability to self-assemble into VLPs, which form the basis of current highly effective HPV vaccines [18]. However, research into next-generation vaccines is increasingly focusing on the L2 protein. L2 offers several key advantages: it can anchor and enhance the stability of L1 complexes, and, more importantly, it exhibits high conservation across various HPV types. This high conservation is crucial because it suggests that L2 could induce cross-immunity against a broader range of HPV strains, addressing a limitation of current type-specific L1-only vaccines [19,20]. Despite ongoing research into the precise durability of L2-induced immune responses, the existing literature strongly supports L2 as a promising target for developing more comprehensive HPV vaccines [19,21,22,23]. This critical feature has directly influenced the development of L1/L2 VLP vaccines. These innovative vaccines aim to provide broader and longer-lasting immunity against a wider spectrum of HPV types, including those not targeted by current L1-based vaccines. Furthermore, the inclusion of L2 opens up the exciting possibility of developing pan-HPV vaccines that offer universal protection and significantly reduce the risk of vaccine escape mutants by targeting conserved epitopes less prone to viral evasion [24,25].

HPV virus-like particles emerge through self-assembly of viral components within the infected cell, persisting through their extracellular stage until they are recognized and infect a new host cell. At this point, they cease to exist as their physical integrity is lost, initiating a new infectious cycle [16]. Techniques for studying the structure of virus particles and components, and some applications of structure-based virus studies, will be presented.

## 2. Overview of Basic Concepts

### 2.1. Virus-like Particle and the Versatility of the Host System of Expression

VLPs mimic the structure of viruses but do not contain genetic material, making them safe for humans and veterinary use (Figure 2). This characteristic enables VLPs to induce a robust immune response, similar to that elicited by real viral infections, without the risk of causing disease [2]. The versatility of this platform allows for its use in various vaccine formulations, targeting viruses, bacteria, and even cancer cells. Another key advantage is its flexible production process, as VLPs can be generated in a variety of host systems, including bacteria, yeast, insect cells, and mammalian cells [8]. Examples of cell systems used to express VLPs include *Escherichia coli* [26,27], *Lactobacillus casei* [1,28], *Pichia pastoris* [29], insect cells [30], and mammalian cell lines [31,32], allowing for scalability and adaptation to the specific needs of each vaccine (Figure 3).

Different research groups have explored a diversity of expression systems. For example, Sakauchi et al. (2021) [32] demonstrated improved scalability by using suspension culture of the human embryonic kidney cell line (HEK 293F) as an expression system for producing the chimeric protein that forms L1/L2 VLPs of HPV16. This approach yielded a higher protein output than expression in HEK 293T, an adherent cell line [32]. Additionally, a lactose-inducible system based on the *Lactobacillus casei* lactose operon promoter has proven effective for HPV16 L1 protein expression in *L. casei*, with successful production of L1 VLPs. This opens new possibilities for developing live mucosal prophylactic vaccines using lactobacilli as delivery vehicles [1,28]. Bei et al. (2024) demonstrated an expression platform utilizing SF9 insect cells (*Spodoptera frugiperda*—fall armyworm, cell line), transfected with baculovirus, as an effective system for producing VLPs that make up the recombinant 14-valent HPV vaccine, currently in phase III trials [30]. Other cell lineages are also utilized as expression platforms for vaccine production. For instance, the Cervarix vaccine was produced by expressing L1-VLPs in the *Trichoplusia ni* (Hi-5) insect cell line [33,34]. Likewise, the yeast species *Saccharomyces cerevisiae* [34] and *Pichia pastoris* [1,29] serve as host systems for manufacturing the Gardasil [35] and WalrinVax [36] vaccines, respectively. Additionally, the bacterium *Escherichia coli* acts as the expression system for producing the Cecolin vaccine [37].

### 2.2. VLPs as a Platform for Drug Delivery

Virus-like particles (VLPs) offer a highly effective drug delivery platform, leveraging their natural ability to penetrate cells—a trait inherited from their viral origins [38]. This intrinsic property enables VLPs to act as efficient nanocarriers, delivering therapeutic cargo directly to target cells, thereby enhancing treatment efficacy while minimizing side effects [39]. The hollow interior of VLPs can encapsulate diverse contents, including nucleic acids, peptides, proteins, and small molecules, making them versatile tools for gene therapy and targeted drug delivery [40]. Additionally, genetic engineering allows the design of scaffold proteins that anchor specific molecules to VLPs. These scaffold proteins can be expressed alongside or separately from the capsid proteins, and upon VLP assembly, they can be displayed externally or embedded internally [41]. Moreover, the protein-based structure of VLP capsids permits extensive chemical modifications. This flexibility facilitates the introduction of functional groups, reversible binding sites, and targeting ligands, further expanding their therapeutic potential [42] (Figure 4).

VLPs can deliver bioactive molecules to modulate gene expression, either activating or inactivating target genes. For instance, Wei and colleagues (2009) demonstrated that RNA encapsulated in MS2 bacteriophage VLPs effectively inhibited hepatitis C virus translation [43]. Beyond gene regulation, VLPs show great promise in anticancer drug delivery. Their unique structure enhances targeted delivery to tumor cells while protecting the therapeutic contents. This approach not only prolongs the drug’s half-life in circulation but also improves cellular uptake, thereby increasing efficacy and reducing off-target effects [42,44]. Studies show that hygromycin conjugated to chimeric M13 phage VLPs results in 1000-fold higher levels of drug delivery in breast cancer cell lines compared to the drug alone [45]. Another impressive case involves *Macrobrachium rosenbergii nodavirus* VLPs loaded with doxorubicin for hyperthermia therapy in cancer cells. This system utilizes covalently decorated folic acid to target a specific receptor on the HT29 cell line, resulting in an increase in doxorubicin cytotoxicity and enhanced drug release at hyperthermia temperatures of approximately 43 °C [46].

In summary, VLPs offer a wide array of applications, presenting numerous advantages as outlined below.

#### 2.2.1. Uses of VLPs

Preventive Vaccines: VLPs have been incorporated into vaccines against viruses such as HPV and Hepatitis B, inducing an effective immune response that helps prevent infections.Oncology Therapies: In addition to vaccines, VLPs are being explored as nanocarriers for cancer therapies, allowing targeted delivery of drugs directly to tumor cells.Personalized Vaccines: VLPs’ versatility enables them to be adapted to create personalized vaccines for various pathogens or virus variants, allowing for a rapid response to emerging outbreaks.Delivery Platforms: VLPs can deliver antigens from different pathogens, facilitating the creation of combination vaccines.

#### 2.2.2. Advantages of VLPs Compared to Other Vaccine Approaches

Safety: Because VLPs do not contain viral genetic material, they cannot cause infections, making them a safe option for vaccination.Immunogenicity: VLPs are highly immunogenic, inducing a strong and long-lasting immune response with fewer doses than other vaccines.Versatility: The ability to modify VLPs to include different antigens enables their use in a wide range of diseases, from viral infections to cancer.Efficient Production: VLPs can be produced in mammalian cells, such as HEK cell lines, which are ideal for large-scale production and can quickly be adapted to various antigens.Fewer Side Effects: The non-infectious nature of VLPs makes them less likely to cause side effects than the attenuated virus vaccines.

### 2.3. Structural Basis of HPV Function

Examining the structural composition and fundamental architecture of viruses is essential for a comprehensive understanding of the mechanisms by which they infect host cells and replicate. The following outlines the main components and characteristics of the HPV.

#### 2.3.1. Basic Structure of HPV

I.Capsid: The capsid is a protein coat that surrounds and protects the virus’s genetic material. It is composed of proteins called capsomeres, which are arranged in specific patterns. The shape of the capsid can be icosahedral, helical, cylindrical, or complex, depending on the type of virus. For example, the capsid of HPV is composed of two structural proteins, L1 and L2, which assemble to form an icosahedral structure. The major capsid protein, L1, organizes into 72 pentameric capsomeres, arranged in a T = 7 symmetric lattice (Figure 5A). These capsomeres create the outer shell of the capsid, providing the virus with its characteristic spherical shape. The minor capsid protein, L2, is located internally and plays a role in stabilizing the structure and facilitating viral genome packaging. Together, L1 and L2 form a robust and highly organized capsid that protects the viral DNA and mediates host cell entry.II.Genetic Material: Viruses contain genetic material, either DNA or RNA, which can be single- or double-stranded. This material allows the virus to hijack host cell machinery, replicate, and produce new viral particles. HPV has a double-stranded DNA genome that encodes proteins for replication and assembly, enabling it to infect and spread within host cells. This genetic diversity is key to viruses’ ability to infect a wide range of hosts and cause disease.

HPV is a non-enveloped virus; some viruses have a lipid envelope surrounding the capsid, which is derived from the host cell membrane. This envelope contains viral proteins that facilitate the virus’s adherence to and entry into cells. Enveloped viruses, such as HIV and influenza viruses, are generally more sensitive to disinfectants and environmental conditions.

#### 2.3.2. HPV Mechanisms of Pathogenesis

I.Attachment and Entry: Proteins on the surface of the viral capsid bind to specific receptors on the host epithelial cell surface—undifferentiated basal cells of stratified epithelia. The HPV major capsid protein L1 interacts with heparan sulfate proteoglycans (HSPGs) on the cell surface. This binding triggers structural changes in the virus, exposing the minor capsid protein L2 after the L1-HSPG interaction. The exposed L2 protein then interacts with additional cell receptors, such as integrins, facilitating clathrin- or caveolin-mediated endocytosis. This process enables the virus to enter the host cell and initiate infection.II.Replication: Once inside a cell, the virus releases its genetic material and uses the host cell’s machinery to replicate itself. HPV’s replication strategy first establishes a low-copy-number replication phase, during which a limited set of *Early* genes (*E1* and *E2*) are expressed. In this stage, the viral genome replicates in sync with the host cell’s DNA, usually just once per cell cycle. This ensures the viral episomes are efficiently maintained and passed to daughter cells as they divide. This quiet phase in the basal layer contributes to the persistent nature of HPV infections, as the low level of viral protein expression minimizes immune detection.III.As infected basal cells begin to differentiate and migrate into the suprabasal layers of the epithelium, an upregulation of viral *Early* gene expression is triggered. This leads to a dramatic viral DNA amplification—known as vegetative replication, with copies reaching hundreds to thousands per cell. This amplification is often achieved through mechanisms that modify the host cell cycle and DNA damage response pathways.IV.Assembly: Simultaneously with the vegetative replication, the *Late* genes (L1 and L2), encoding the viral capsid proteins, are expressed. The newly synthesized viral DNA is then encapsulated into mature virions within the nucleus of these differentiating cells.V.Release: Finally, the new HPV virions are released from the host cell, completing the viral life cycle and enabling further transmission.

In conclusion, the structural basis of viral function and the basic architecture of viruses are fundamental to understanding how these infectious agents operate and replicate. The capsid, composed of proteins, protects the virus’s genetic material and is essential for its replication and infection [2].

These structures not only define the shape and stability of viruses but also determine their ability to infect cells and spread. The interaction between viral proteins and cellular receptors is crucial for virus adhesion and entry, while replication and assembly depend on the host cellular machinery [47].

To summarize, viral architecture is a crucial determinant of the efficacy of infection and the pathogenicity of viruses. Understanding these structures and functions is essential for the development of vaccines and antiviral therapies, allowing the creation of effective strategies to combat viral infections and protect public health [47,48].

## 3. VLP Vaccines and the Role of Capsid and Other HPV Proteins in Research

Cervical cancer is the fourth most prevalent type of cancer among women worldwide, with approximately 600,000 new cases registered in 2020. Data indicates that approximately 90% of the 342,000 deaths recorded from this type of cancer occurred in low-income countries [49]. There is also an association between HPV and other types of cancer, such as head, neck, and oropharyngeal carcinomas [50,51,52]; anogenital (vaginal, vulvar, penile, and anal) [53,54]; and cases of non-melanoma skin cancers [55,56].

The GLOBOCAN 2022 database provides estimates of new cancer cases and deaths worldwide for all cancers combined (ICD-10 codes C00-C97) and for 36 specific cancer types. These include oropharynx, anus, vulva, vagina, cervix uteri, penile, and nonmelanoma skin cancer (NMSC; ICD-10 code C44, excluding basal-cell carcinomas) [57]. The global cancer burden for 2022, based on the latest GLOBOCAN estimates by the International Agency for Research on Cancer (IARC), is accessible through the Global Cancer Observatory’s “Cancer Today” platform [58].

Currently, the primary method of preventing cervical cancer is through periodic cytological exams, Pap smear tests, and prophylactic vaccination of pre-adolescent girls and boys. In Brazil, the prophylactic vaccine Gardasil^®^ quadrivalent was made available by the Ministry of Health (Ministério da Saúde do Brasil), free of charge through the Unified Health System (Sistema Único de Saúde—SUS) for boys and girls aged 9 to 19 [59], as well as for men and women aged 9 to 26 living with HIV or AIDS, patients who have received organ transplants or bone marrow transplants, and people undergoing cancer treatment [60]. Individuals outside the age range covered by the SUS can receive HPV immunization at private vaccination clinics, with two vaccine options available: the quadrivalent vaccine, recommended for females aged 9 to 45 and males aged 9 to 26, and the nonavalent vaccine, suitable for everyone aged 9 to 45.

The vaccines can be used by people who are undergoing treatment or have already had an HPV infection, as they can protect against other types of HPV and prevent the formation of new genital warts and the risk of cancer [61]. Currently, the 9-valent Gardasil^®^ is the most effective vaccine, able to promote immunization against nine HR-HPVs (HPV 6, 11, 16, 18, 31, 33, 45, 52, and 58). Table 1 provides a comparative analysis of commercially approved HPV vaccines, detailing the protective strains, expression systems, and adjuvants utilized. Adjuvants are essential components of HPV vaccines, as they enhance immunogenicity and ensure sustained protection. Gardasil vaccines incorporate amorphous aluminum hydroxyphosphate sulfate (AAHS), a conventional aluminum salt adjuvant, to extend antigen persistence and facilitate phagocytic uptake by antigen-presenting cells, thereby eliciting a robust Th2-mediated antibody response [62,63]. In contrast, Cervarix employs AS04, an advanced adjuvant system that combines monophosphoryl lipid A (MPL)—a Toll-like receptor 4 (TLR4) agonist—with aluminum hydroxide [64]. This synergistic formulation not only augments humoral immunity but also induces a Th1-polarized response, thereby enhancing cellular immune activation [64]. Additionally, aluminum hydroxide and aluminum phosphate are traditional adjuvants used in various formulations, promoting antigen depot formation and dendritic cell recruitment [65].

Nevertheless, other options have been studied, such as a 9-valent HPV vaccine in development by Shanghai Bovax, which targets the same HPV types as 9-Gardasil^®^, showing similar immunogenicity, tolerance, and safety against HPV types 6, 11, 16, and 18, and demonstrated non-inferiority when compared to 4-Gardasil^®^ [66]. In turn, the recombinant 14-valent HR-HPV vaccine (SCT 1000) offers protection against the same strains as 9-Gardasil^®^, in addition to five other strains (HPV-35, 39, 51, 56, and 59) and is in phase III trials in China [30].

### 3.1. Chimeras Designed for New HPV Vaccines

VLPs were recognized as a new generation of prophylactic vaccines against viral infections two decades ago. Today, they serve as the foundation for both prophylactic nanovaccines and therapeutic drug nanocarriers, which are often constructed as multivalent chimeric VLPs (chi-VLPs) through genetic modifications or chemical conjugation. Continuous advancements in genetic engineering have expanded the strategies available for developing multivalent chi-VLP vaccines. This section highlights studies focused on chimeric capsid VLPs and chimeric enveloped VLPs, discussing the advantages and limitations of each approach [67].

Key requirements for designing multivalent chi-VLP vaccines include the following:Surface-exposed antigens must be incorporated without compromising the structural stability of the chi-VLPs.The chi-VLPs should induce protective antibodies against both the displayed antigen and the source virus of the VLPs.

Notably, antigenicity is not a prerequisite for VLPs that function solely as carriers for antigen display or drug delivery [67].

Currently, three licensed vaccines are based on the mixture of L1 VLPs from two, four, or nine different HR-HPVs [68]. Rather than increasing the diversity of L1-based VLPs, emerging strategies focus on developing prophylactic vaccines using chi-VLPs. These VLPs are formed by recombinant proteins that incorporate highly conserved neutralizing epitopes from the structural protein L2, inserted into the L1 protein. Studies, such as those conducted by Boxus et al. (2016) [15], have demonstrated that chimeric L1/L2 HPV-based VLPs (chi-L1/L2 VLPs) can effectively display selected L2 epitopes at doses comparable to those of licensed vaccines. Furthermore, these chi-VLPs induce robust immune and protective responses against multiple HR-HPV types [15]. Such findings suggest that chi-L1/L2 VLPs represent a promising, cost-effective approach for developing broadly protective vaccines against various HPV types.

The minor capsid protein L2 harbors highly conserved, HPV-neutralizing epitopes that are shared across multiple high-risk HPV types [69]. Recent research has explored the potential of broadening vaccine-mediated protection through AS04-adjuvanted vaccines based on chi-L1/L2 constructs. These chi-VLPs were designed by integrating L2 epitopes into the DE loop and/or C-terminus of the L1 protein. One such construct, an HPV18 L1-based chi-VLP, incorporated HPV33 L2 epitopes (amino acid residues 17–36) into the L1 DE loop and HPV58 L2 epitopes (amino acid residues 56–75) into the L1 C-terminus. This chi-VLP elicited durable immune responses and demonstrated cross-protection against multiple HPV types, including HPV6, 11, 16, 31, 35, 39, 45, 58, and 59, as evidenced by pseudovirion and quasivirion assays in mouse and rabbit models. In rabbits, the breadth and magnitude of protection were further amplified when the chi-L1/L2 VLPs were administered alongside HPV16/18 L1 VLPs [15]. These findings highlight the potential of the HPV18 L1/L2 chi-VLP, particularly when formulated with AS04 and combined with HPV16/18 L1 VLPs, as a promising strategy for achieving broad-spectrum protection against diverse HPV types in humans [6].

Personalized cancer immunotherapies, combined with nanotechnology (nanovaccines), are revolutionizing treatment strategies for HPV-related cancers, complementing advancements in chi-HPV vaccine studies. Chi-HPV vaccines, designed to incorporate L1 and L2 epitopes from multiple HPV types, aim to broaden the protective immune response and prevent a broader range of HPV infections [13,15]. While these vaccines address prevention, personalized nanovaccines focus on treating existing cancers by tailoring therapies to the unique genetic and molecular profiles of individual tumors [6].

Nanotechnology plays a pivotal role in both approaches. In chi-HPV vaccines, nanoparticles enhance the stability and immunogenicity of VLPs, improving their ability to elicit broad and durable immune responses. Similarly, in personalized cancer vaccines, nanocarriers such as polymeric nanoparticles and high-density lipoprotein-mimicking nano-discs facilitate efficient delivery of tumor-specific antigens, boosting immune activation and precision in targeting HPV-related malignancies [70].

Recent preclinical studies have highlighted the potential of chi-L1/L2 VLPs to provide cross-protection against diverse HPV types, extending beyond those targeted by current prophylactic vaccines. Personalized cancer nanovaccines build on this progress by addressing therapeutic gaps and enabling the treatment of HPV-related cancers, such as cervical and oropharyngeal cancers. Additionally, integrating proteolysis-targeting chimeras (PROTACs) with nanotechnology offers a novel approach to selectively degrade oncogenic HPV proteins, enhancing treatment efficacy. The synergy between chi-HPV vaccines, personalized nanovaccines, and advanced technologies like PROTACs presents a comprehensive strategy to combat HPV-related diseases, overcoming challenges in nanocarrier optimization and safety to provide innovative solutions for both prevention and treatment [71].

### 3.2. Some Examples of Chi-VLP Applications

Qi et al. (2022) [72] constructed a novel multi-epitope vaccine of HPV16 E5E6E7 oncoprotein delivered by HBc VLPs, which induced efficient prophylactic and therapeutic antitumor immunity in a tumor mouse model. The E5aa28-46, E6aa37-57, and E7aa26-57 peptides were selected and linked to form a novel multi-epitope vaccine (E765m), which was inserted into the major immune-dominant region (MIR) of hepatitis B virus core antigen (HBc) to construct an HBc-E765m chimeric virus-like particle (chi-VLPs). The immunogenicity and immunotherapeutic effect of the chi-VLPs vaccine were evaluated in immunized mice and a tumor-bearing mouse model. The results showed that HBc-E765m chi-VLPs elicited high E5-, E6-, and E7-specific CTL and serum IgG antibody responses, and relatively high levels of the cytokines IFN-γ, IL-4, and IL-5. More importantly, the chi-VLPs vaccine significantly suppressed tumor growth in mice bearing E5-TC-1 tumors. Our findings provide strong evidence that this novel HBc-E765m chi-VLPs vaccine could be a candidate vaccine for specific immunotherapy in HPV16-associated cervical intraepithelial neoplasia or cervical cancer [72].

Zhou and Zhang (2023) [73] developed a chimeric VLP vaccine against pulmonary tuberculosis. Bacillus Calmette-Guérin (BCG), the only current vaccine against tuberculosis (TB) that is licensed in clinics, successfully protects infants and young children against several types of TB, such as TB meningitis and miliary TB. Still, it is ineffective in protecting adolescents and adults against pulmonary TB. Thus, it is a matter of the highest urgency to develop an improved and efficient TB vaccine. In this background, VLPs exhibit excellent characteristics in the field of vaccine development due to their numerous factors, including but not limited to their good safety without the risk of infection, their ability to mimic the size and structure of original viruses, and their ability to display foreign antigens on their surface to enhance the immune response. In this study, the HPV16 L1 capsid protein (HPV16L1) acted as a structural vaccine scaffold, and the extracellular domain of Ag85B was selected as the M. tb immunogen and inserted into the FG loop of the HPV16 L1 protein to construct chimeric (chi-) HPV16L1/Ag85B VLPs. The chi-HPV16L1/Ag85B VLPs were produced via the *Pichia pastoris* expression system and purified via discontinuous Optiprep density gradient centrifugation. The humoral and T cell-mediated immune response induced by the chi-HPV16L1/Ag85B VLP was studied in female C57BL/c mice. In this context, it was demonstrated that the insertion of the extracellular domain of Ag85B into the FG loop of HPV16L1 (Figure 5B) did not affect the in vitro stability and self-assembly of the chi-HPV16L1/Ag85B VLPs [22].

Notably, it did not interfere with the immunogenicity of Ag85B. The chi-HPV16L1/Ag85B VLPs were observed to induce higher Ag85B-specific antibody responses and elicit significant Ag85B-specific T cell immune responses in female C57BL/6 mice compared with recombinant Ag85B. These findings provide new insights into the development of novel chi-HPV16L1/TB VLP-based vaccine platforms for controlling TB infection, which are urgently required in low-income and developing countries [73].

Chen et al. (2022) [74] technologically advanced on the construction of the HIV-1 P18I10 CTL peptide derived from the V3 loop of HIV-1 gp120, and the T20 anti-fusion peptide of HIV-1 gp41 was inserted into the HPV16 L1 capsid protein to construct chi-HPV:HIV (L1:P18I10 and L1:T20) VLPs by using the mammalian cell expression system. The HPV:HIV VLPs were purified by chromatography. It was demonstrated that the insertion of P18I10 or T20 peptides into the DE loop of HPV16 L1 capsid proteins did not affect in vitro stability and self-assembly or the morphology of chi-HPV:HIV VLPs.

Notably, it did not interfere either with the HIV-1 antibody reactivity targeting sequential and conformational P18I10 and T20 peptides presented on chi-HPV:HIV VLPs or with the induction of HPV16 L1-specific antibodies in vivo. It was possible to observe that chi-L1:P18I10/L1:T20 VLPs vaccines could induce HPV16- but weak HIV-1-specific antibody responses and elicited HPV16- and HIV-1-specific T-cell responses in BALB/c mice [74].

Moreover, an additional dose of the immunizing agent could be a potential booster, increasing or renewing the effect of the earlier dose, thereby enhancing HIV-specific cellular responses in the heterologous immunization after priming with the rBCG HIV vaccine. This research work would contribute a step towards the development of the novel chi-HPV:HIV VLP-based vaccine platform for controlling HPV16 and HIV-1 infection, which is urgently needed in developing and industrialized countries [74].

Chen et al. (2023) [75] considered that human papillomavirus (HPV) vaccines based on HPV L1 virus-like particles (VLPs) are already licensed but not accessible worldwide, and that about 38.0 million people were living with HIV in 2020 and that there is no HIV vaccine yet; therefore, safe, effective, and affordable vaccines against both viruses are an urgent need. Therefore, the improvement of the construction of the HIV-1 P18I10 CTL peptide from the V3 loop of the HIV-1 gp120 glycoprotein was inserted into the HPV16 L1 protein to construct chi-HPV:HIV (L1:P18I10) VLPs. Instead of the traditional baculovirus expression vector/insect cell (BEVS/IC) system, an alternative mammalian 293F cell-based expression system was established using cost-effective polyethylenimine-mediated transfection for L1:P18I10 protein production. Compared with conventional ultracentrifugation, a novel chromatographic purification method, which could significantly increase L1:P18I10 VLP recovery (~56%), was optimized.

Chi-L1:P18I10 VLPs purified from both methods were capable of self-assembling into integral particles and shared similar biophysical and morphological properties. After BALB/c mice immunization with 293F cell-derived and chromatography-purified L1:P18I10 VLPs, almost the exact titer of anti-L1 IgG (*p* = 0.6409) was observed as in Gardasil anti-HPV vaccine-immunized mice. Significant titers of anti-P18I10 binding antibodies (*p* < 0.01%) and P18I10-specific IFN-γ secreting splenocytes (*p* = 0.0002) were detected in L1:P18I10 VLP-immunized mice in comparison with the licensed Gardasil-9 HPV vaccine. Furthermore, it was demonstrated that insertion of the HIV-1 P18I10 peptide into the HPV16 L1 capsid protein did not affect the induction of anti-L1 antibodies. Altogether, it was expected that the mammalian cell expression system and chromatographic purification methods could be time-saving, cost-effective, scalable platforms to engineer bivalent VLP-based vaccines against HPV and HIV-1 [75].

Adjusting the effective and safe dose for HPV chi-VLPs is crucial. Administering a higher dose of HPV chi-VLPs can lead to two possible outcomes: while it may trigger a more robust immune response, it risks intensifying adverse immunological effects [67]. At increased doses, common transient side effects like injection site pain, swelling, and systemic reactions such as headache and fever could become more pronounced [76]. More concerningly, a significantly higher antigen load could theoretically overstimulate the immune system, potentially leading to increased systemic inflammation or, in rare instances, more severe allergic or hypersensitivity reactions [77]. While current HPV VLP vaccines have an excellent safety record at standard doses, a review of the literature reveals that severe adverse events, though extremely rare, have been reported in the context of vaccine use, including instances of anaphylaxis, syncope, and, very rarely, autoimmune-like conditions, although direct causality to VLP dose has not been definitively established for these rarer events [62,66,78]. For example, some case reports in the literature have described occurrences of conditions like Guillain–Barré syndrome or acute disseminated encephalomyelitis following HPV vaccination, though large-scale studies have generally not found a causal link, and these remain extremely rare occurrences that are not dose-dependent in the established literature [79].

To mitigate these potential adverse effects, a multifaceted strategy is needed. Rigorous preclinical and dose-escalation clinical trials are crucial to determining the optimal VLP dose that balances strong immunogenicity with minimal reactogenicity. The development of advanced adjuvant systems that precisely modulate the immune response, rather than simply amplifying it, could allow for lower effective VLP doses. This could involve adjuvants that selectively promote desired immune pathways or increase the efficiency of antigen presentation. Furthermore, optimizing the design of the chi-VLPs themselves so that they are more efficiently recognized and processed by immune cells could reduce the required amount of antigen. Ultimately, the goal is to achieve maximal protective immunity with the lowest possible VLP dose, ensuring high efficacy and an excellent safety profile, while continuously monitoring for any unanticipated adverse events through robust post-marketing surveillance.

### 3.3. Further Applications of VLPs in Drug Delivery

The immunogenic features of VLP nanoparticles make them attractive for further exploration of their potential abilities, particularly in drug delivery and gene therapy. Beyond transporting peptides/proteins or other active molecules on the surface of the VLPs, they can bind proteins, nucleic acids, or other small molecules. Consequently, they can be used to deliver molecules to specific cells, tissues, or organs [6].

The VLPs’ uptake by the cells occurs via receptor-mediated endocytosis through the plasma membrane that surrounds VLPs, internalizing them into the cell as a vesicle. So, after the vesicles detach from the membrane, they arrive in the cytosol. Then, vesicles are moved along the cytoskeleton for association with the primary endosomes. Sometimes, the endosomal vesicles separate from the primary endosome, grow up as the final endosome, and fuse with the pre-lysosomal vesicles containing acidic hydrolase to form lysosomes. The unknown materials are disintegrated inside the lysosomes and made available to the cells. Nevertheless, lysosomal degradation is also a problem because it prevents proper drug delivery, due to about 40% of newly produced drugs being disapproved as a result of poor bioavailability [6].

The use of drug nanocarriers is an advantageous strategy to solve this limitation. Until now, several specific sites, e.g., tumor tissues, have reduced the required dose of the drug and improved treatment results. Interestingly, some VLPs present a natural tropism toward a particular tissue, due to the virus from which they originated. For example, HBV naturally infects the liver so that HBV-derived VLPs can target liver cells. In the same way, rotaviruses have a special attraction for the intestine, and their derived VLPs can be used for drug delivery to the intestinal tissue [6].

VLPs are preferably taken up by tumor cells due to their ability to be engineered with receptor-binding domains on their surface. These domains, which can be chemically or genetically attached, allow VLPs to selectively bind to specific receptors highly expressed on cancer cells, thus enhancing the targeted delivery of drugs and improving therapeutic effects [80]. Additionally, some VLPs retain a natural tropism towards certain tissues, inherited from their parental viruses; for example, HBV VLPs present liver tropism [81] and rotavirus VLPs exhibit a specific tropism targeting the differentiated enterocytes of the small intestine’s villi [82]. This feature can also contribute to their selective uptake by specific cells, including those in tumors or other diseased tissues.

More specific targeting is typically achieved by arranging the receptor-binding domain on the VLP surface. Target domains can be chemically or genetically attached to the surface of VLPs, allowing the VLPs to selectively bind to cancer cells expressing a specific receptor, enhancing the therapeutic effects of drugs [83]. VLPs can also deliver nucleic acids. For example, it has been demonstrated that the systematic delivery of a known gene silencer, miR-146a, via bacteriophage MS2-derived VLPs is an effective treatment for reducing inflammatory cytokines in mice carrying systemic lupus erythematosus [83]. Examples of practical uses of VLPs for drug delivery are shown in Table 2, and VLP-based vaccines against distinct cancers are shown in Table 3.

VLPs elicit strong immune responses across different immune cell subsets. They are readily recognized by antigen-presenting cells (APCs) like dendritic cells, leading to their maturation and subsequent activation of T helper (CD4+) and cytotoxic T (CD8+) lymphocytes [118]. Furthermore, VLPs can directly activate B cells, stimulating robust antibody production. These immune responses should generally not be avoided, as their intrinsic immunogenicity can be highly beneficial for drug delivery. The strong immune activation serves as a natural adjuvant effect, enhancing the overall immune response to the co-delivered therapeutic cargo, which is particularly advantageous for vaccine applications or immunotherapies [80]. Moreover, for cancer drug delivery, the immune response against the VLP itself, or against tumor antigens presented by engineered VLPs, can contribute to an anti-tumor effect, synergistically benefiting the treatment [119]. While a pre-existing anti-VLP immune response could theoretically lead to faster clearance upon repeated administration, strategies like using VLPs from non-human viruses or modifying their surface can mitigate this, ensuring the generated immune responses are primarily leveraged for enhanced therapeutic outcomes and targeted drug delivery [118,119].

### 3.4. Other Applications Based on HPV Proteins

Despite the availability of prophylactic vaccines, HPV infections remain highly prevalent worldwide [120,121]. There is no specific treatment for HPV infections or lesions that have already developed, making prophylactic vaccination and preventive measures essential for reducing infection risk and detecting HPV-related lesions early. Among the proteins expressed by HPV, L2, E7, and E6 are prominent targets for the development of DNA vaccines, a plasmid-based vector technology for gene delivery with antigenic potential, making it a promising alternative to traditional platforms due to its simplicity, low production costs, thermal stability (which eliminates the need for refrigeration), and the ability to be administered in repeated doses without triggering an immune response against the delivery vector [122]. The L2 protein is a component of the viral capsid and presents a high level of sequence conservation across various HPV types, making it a promising candidate for cross-protective vaccines [22]. E6, an oncogenic protein, plays a key role in the malignant transformation of infected cells, particularly in conjunction with E7. The oncogenes *E6* and *E7* are essential for malignancy, as they integrate into the host DNA during cancer progression. E6 promotes the degradation of the tumor suppressor protein p53, while E7 inactivates the retinoblastoma protein (pRb), disrupting cell cycle regulation and enabling uncontrolled cell proliferation [123,124,125].

Proteins such as L2 and E6 can be utilized as promising tools in the development of DNA vaccines, being expressed directly inside cells as recombinant proteins that combine regulatory and coding sequences, thereby eliciting both humoral and cellular immune responses. It has been demonstrated that the L2 protein induces humoral action leading to the production of anti-L2 antibodies with evident prophylactic action, while the E6 protein promotes the expression of tumor necrosis factor (TNF) with high prevalence against malignant cells, generating the possibility of anticancer therapeutic action [126,127].

The VLP structural proteins may contain several epitopes that are searchable by molecular modeling and in silico assays, identifying high-level immune stimulation peptide sequences [128]. These epitopes contain several peptides that could be applied to develop peptide-based vaccines. Peptide epitopes have several advantages in their application as antigens for developing prophylactic and therapeutic vaccines. They can be synthesized either chemically or biologically using several approaches, including solid-phase peptide synthesis (SPPS) [129,130], cell-free systems [131,132], and recombinant production in bacteria, yeast, or mammalian cells [32,132].

Zhai et al. (2017) [105] observed that peptides from HPV L2-protein (amino acid sequences 17–36, 69–86, and 108–122) could be coated on MS2 bacteriophage VLP to provide complete and cross-protection against HR-HPV types, protecting mice from cervicovaginal infections with HPV pseudoviruses 16, 18, 31, 33, 45, and 58 at levels comparable to those immunized with Gardasil-9 [105]. In turn, Namvar et al. (2019) observed highly conserved peptide sequences between HR-HPV subtypes in the L1 (five regions) and L2 (four regions) proteins, which showed a high affinity for MHC-I and MHC-II receptors, suggesting the ability to elicit cellular and humoral immune responses [133].

Research into peptide-based vaccines against HPV seeks applied peptide epitopes from structural proteins L1 and L2, as well as the oncoproteins E6 and E7, which are involved in HPV carcinogenesis by interfering with cell cycle control and interacting with regulatory proteins. E6 interacts with the ubiquitin-protein ligase E3A, known as E6AP, forming a complex responsible for the degradation of tumor protein p53, resulting in the loss of the cell cycle checkpoint and evading apoptosis [134]. In turn, E7 triggers the release of the transcription factor E2F and the degradation of retinoblastoma protein (pRb), driving cells to enter S-phase (synthesis phase of the cell cycle for DNA replication) prematurely through the transcription of cyclins and tumor suppressor protein p16INK4A [128,135]. Utilizing E6 and E7 epitopes presents a promising approach for developing therapeutic vaccines, offering potential treatment for pre-existing infections and established tumors, as well as generating protection against new infections [136].

Both DNA and peptide vaccine platforms, as well as VLP technology, have benefited from bioinformatic studies. Computational techniques, including Computer-Aided Design and Drafting (CADD) and in silico assays, are essential tools in HPV vaccine development. The main advantage of CADD is that it allows modeled ideas to be explored before physical prototyping is implemented. CADD streamlines the design of multiepitope constructs by predicting interactions between HPV antigens (e.g., L1, L2, E6, and E7) and immune receptors, while in silico assays simulate immunogenicity, toxicity, and structural stability. These methods reduce costs and accelerate candidate selection. However, their reliance on simplified models and incomplete databases requires experimental validation to confirm predictions [137,138].

Diverse treatments have already been tested to combat HPV-associated diseases, including the use of therapeutic vaccines developed on different platforms, such as attenuated vectors (viral or bacterial), peptide/protein vaccines, and even cell-based vaccines [133,139,140,141]. Several HPV vaccines with therapeutic potential are in advanced stages of clinical trials. These vaccines utilize various epitopes derived from the oncoproteins E6 and E7, as well as the structural proteins L1 and L2, and are developed on diverse vaccine platforms [139]. Notable examples include HPV16 L1E7 chimeric virus-like particles (CVLPs) [76], the ZYC-101a DNA vaccine (encoding HPV16/18 E6 and E7) [142], the TA-CIN protein vaccine (an E6-E7-L2 tandem fusion protein) [143], TA-HPV (a recombinant vaccinia virus expressing E6 and E7) [144], and the SGN-00101 protein vaccine (a fusion of heat shock protein with HPV16 E7) [145]. While each approach offers distinct advantages, they also present certain limitations. Consequently, identifying potent and safe strategies to enhance the immunogenicity of therapeutic vaccines remains a critical priority. In this context, the use of computational tools, such as in silico studies, is essential for advancing vaccine development [141,146,147].

At this point, immunoinformatic tools can assist scientists in predicting high-immunogenic and conserved epitopes that induce B- or T-cell responses against HPV infection. The heat shock proteins (HSPs) are promising adjuvants able to stimulate innate and adaptive immunity, making the linkage of antigens to HSPs a promising strategy to increase the efficiency of vaccine candidates [141]. In silico approaches were employed to design the multiepitope and HSP-epitope linkage, such as L1-L2-E7 and HSP70-L1-L2-E7 constructs, which are promising vaccine candidates. The E7 protein, used alone or in combination with other HPV proteins, particularly E6, has been a focus in the development of therapeutic clinical trials [147]. Moreover, vaccine constructs incorporating the E7 protein along with L1 and L2 proteins were evaluated. The expression of both multiepitope DNA constructs was analyzed in a mammalian cell line. Finally, their immunological and anti-tumor effects were assessed in C57BL/6 mice, demonstrating promising results [146].

While computational approaches cannot fully replace experimental testing, they provide a robust framework to enhance the efficiency and precision of HPV vaccine development, bridging gaps between design and real-world application, as well as accelerating the research steps [137,138,148].

## 4. A Brief Overview of Methods for Studying Viral Particles

The detailed analysis of viral structure and physical properties depends on a range of advanced scientific techniques [149,150]. Imaging methods such as electron microscopy (Figure 6A,B) and X-ray crystallography deliver high-resolution views of viral shapes and atom-level protein arrangements, respectively. Spectroscopic techniques like nuclear magnetic resonance spectroscopy and mass spectrometry are essential for examining protein structures in solution and determining protein composition and modifications [151,152]. Biochemical analyses, including virus purification and stability tests, help define physical characteristics like size, shape, and environmental resistance [153]. Supporting these empirical techniques, computational modeling, especially molecular simulations (Figure 5A,B), provides predictive insights into protein structure and interactions, clarifying the relationship between structure, function, and pathogenicity [152,153].

The integration of genomics and proteomics has significantly advanced viral characterization. Genetic sequencing provides essential information on viral genome organization and encoded proteins, while proteomic analysis is fundamental for identifying and quantifying viral proteins. A key application of proteomics is the functional analysis of protein–protein interactions (PPIs) between viral and host cell components. Understanding these PPIs is vital, as they govern critical aspects of the viral life cycle, immune evasion, and manipulation of host cellular processes, ultimately influencing viral pathogenicity [154]. Advanced techniques within proteomics, including mass spectrometry-based identification, quantification methods, and various functional assays, continue to deepen our understanding of these intricate host–virus relationships, driving progress in virology and the development of new interventions [149,150].

## 5. Conclusions and Future Directions

Based on significant advances in nanotechnology and protein engineering, chimeric L1/L2 HPV-based VLPs (chi-L1/L2 VLPs) represent a highly promising and versatile platform for both innovative vaccines and advanced nanocarriers. Given their self-assembling nature and the absence of viral genetic material, these nanoparticles offer an inherently safe and highly immunogenic approach to combating diverse pathogens.

The ability of chi-L1/L2 VLPs to competently and recurrently present multivalent antigens is a key advantage, facilitating targeted delivery that can induce robust and long-lasting immune responses. This transformation highlights a shift in perspective, where viruses, once viewed solely as threats, now serve as templates for powerful biomedical tools.

Chi-L1/L2 VLPs enable the construction of precise nanostructured arrays, which can be manipulated through the insertion of various epitopes. This allows for the creation of customized polyvalent antigenic nanoparticles to address diverse public health needs. Furthermore, the ability of VLPs to escape endosomes before lysosomal degradation significantly enhances their efficacy as delivery vehicles.

Overall, VLPs offer a compelling set of properties that make them highly advantageous as drug delivery systems. These properties include allowing chemical modifications and drug insertion into their cavity, high immunogenicity, in vivo persistence, safety, and efficient delivery. They are also highly effective at low doses and demonstrate favorable stability under refrigeration, along with a high manufacturing safety profile. Although their production speed cannot yet rival that of mRNA or viral vector vaccines, the proven benefits of VLP technology are undeniable.

In the global effort to prepare for future pandemics, a multiplatform strategy is crucial. This involves continued investment in and integration of diverse platforms, such as mRNA vaccines, viral vector-based vaccines, peptide vaccines, traditional VLPs, and potent chimeric VLPs, like chi-L1/L2 VLPs. This comprehensive approach, backed by international collaboration in vaccine development and distribution, is essential for strengthening global health systems and mitigating the severe social and economic impacts of emerging infectious diseases.

## Figures and Tables

**Figure 1 viruses-17-01209-f001:**
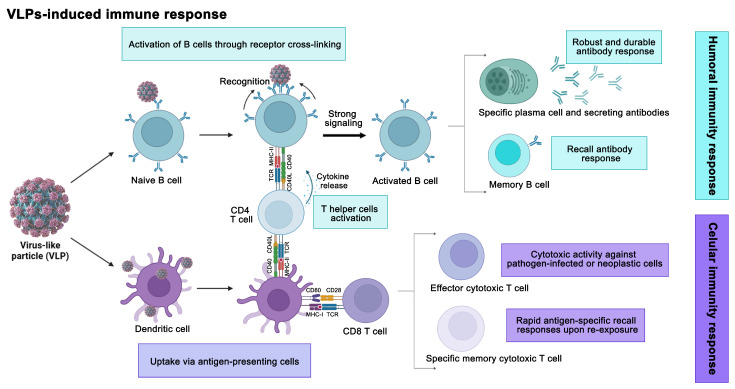
Virus-like particles (VLPs) activate both humoral and cellular immune responses. Dendritic cells take up VLPs and present their antigens via MHC-I and MHC-II, stimulating CD8+ and CD4+ T cells. CD8+ T cells differentiate into cytotoxic effectors and memory cells, providing long-term cellular immunity. Simultaneously, VLPs directly cross-link B cell receptors and, with CD4+ T helper cells (Th), drive plasma cell antibody production and memory B cell formation. This dual response ensures robust, long-lasting protection (VLP AI model by Nierengarten, 2025 [11]).

**Figure 2 viruses-17-01209-f002:**
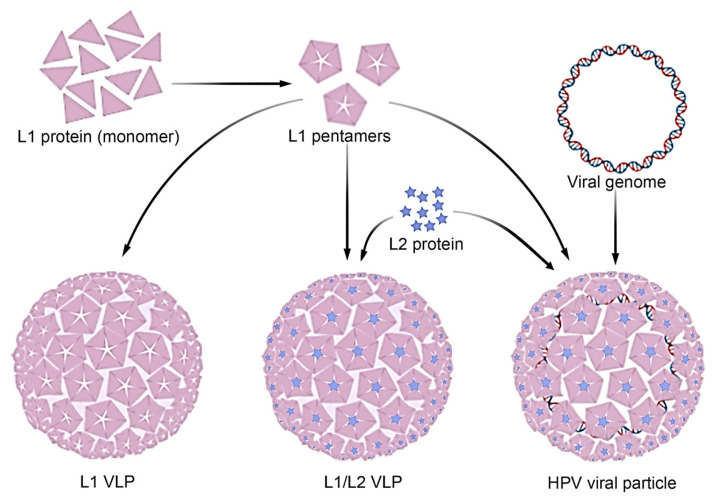
The difference between HPV virus-like particles (VLPs) and the HPV virus is significant. L1 is a structural protein capable of forming organized structures, grouping into 72 pentamers. The second structural protein of HPV is L2, which cannot self-assemble independently. However, L2 can co-assemble with L1, stabilizing L1 pentamers and enabling the formation of chimeric HPVL1/L2 VLPs. The HPV viral particle consists of a structural capsid formed by L1 and L2 proteins, encapsulating the viral genome.

**Figure 3 viruses-17-01209-f003:**
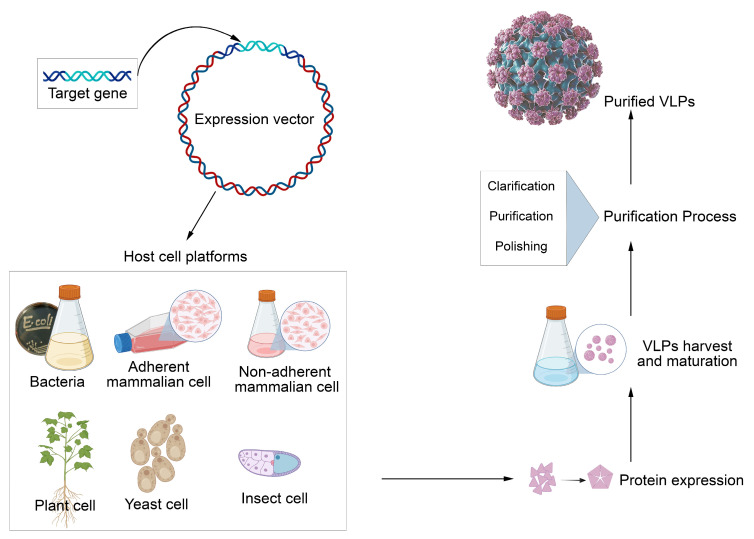
Schematic steps to obtain VLP: First, the target gene can be inserted into an expression vector, and subsequently, the modified vector can be introduced into a host cell. The protein should then be expressed using the chosen host cell platform. Following the expression, the cell products should be processed, followed by VLP maturation and purification (VLP AI model by Nierengarten, 2025 [11]).

**Figure 4 viruses-17-01209-f004:**
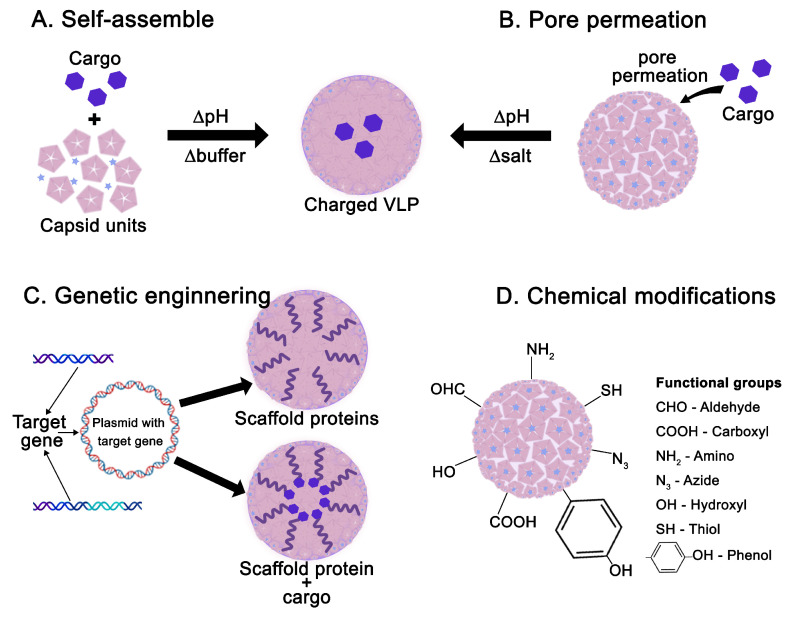
Schematic approaches to VLPs as nanocarriers for various molecules and drugs. (**A**) Self-assembly of capsid units around cargo occurs by altering pH (∆pH) and buffer (∆buffer) conditions, producing a VLP with inner cargo. (**B**) Cargo permeates through pores due to changes in pore diameter promoted by alterations in pH and salt (∆salt) concentration. (**C**) Target genes are inserted into plasmids using genetic engineering techniques to express scaffold proteins, enabling cargo conjugation and encapsulation. (**D**) Chemical modifications on the VLP surface occur through the insertion of different functional groups for tailored applications (adapted by Chung et al., 2020 [41]).

**Figure 5 viruses-17-01209-f005:**
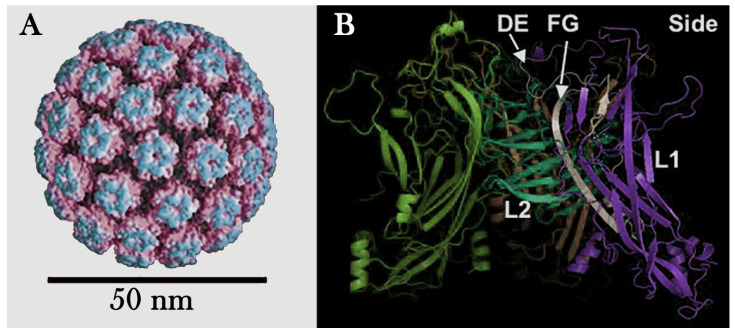
(**A**) Atomic model showing the T = 7 symmetry of the HPV capsid, highlighting the main capsid protein L1 organized into pentameric capsomeres shown in blue (adapted by Zhao et al., 2012 [16]). (**B**) Predicted 3D model of the interaction between L1 and L2 proteins. The HPV16 L1 structure was obtained from the RCSB Protein Data Bank, and the HPV16 L2 protein was predicted using the 3D-Jigsaw and Swiss Model Server. DE and FG correspond to specific interactions predicted between L1 and L2. The DE loop and the FG loop of L1 and specific proline-rich regions of L2 are included. These prolines vary from highly conserved to completely conserved among all alpha-PVs (alpha-papillomaviruses), including all HPV types, based on the protein–protein interaction model of the L1 and L2 monomers. L2 probably binds within the center of the L1 pentamer. The position of the L2 antigenic region, therefore, is predicted to face superficially or externally when bound to the L1 pentamer (adapted by Lowe et al., 2008 [22]).

**Figure 6 viruses-17-01209-f006:**
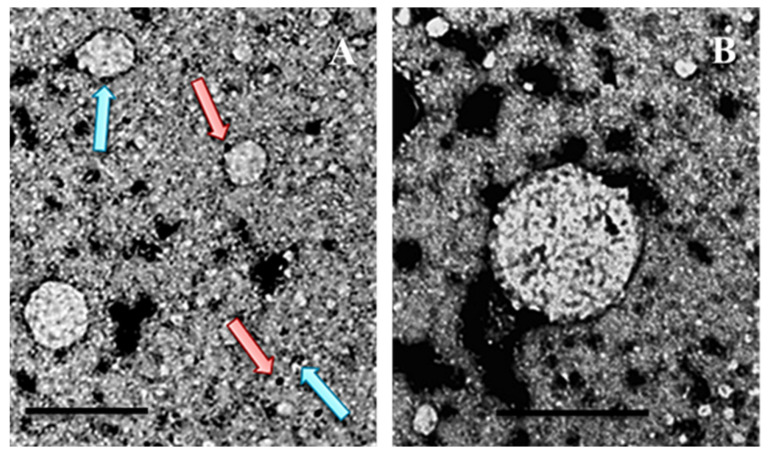
Chimeric L1/L2 VLPs were produced in a large matrix of capsid proteins from HPV16. These proteins were obtained using HEK 293T cells as the host system. (**A**) Transmission electron microscopy images show VLPs in formation, with the larger VLP measuring approximately 30 nm in diameter. Ultrastructural immunocytochemistry detection for L1 and L2 proteins, employing 5 nm (blue arrows) and 10 nm (pink arrows) colloidal gold particles, was applied, respectively. (**B**) A highlight showing the L1/L2 VLP of HPV16 in an advanced maturation stage, measuring around 50 nm in diameter. Bars = 50 nm (Images by Cianciarullo et al. 2024 [1]).

**Table 1 viruses-17-01209-t001:** Comparison between prophylactic-based VLP vaccines against HPV.

	Quadrivalent Gardasil^®^	Bivalent Cervarix^®^	Nonavalent Gardasil-9^®^	Bivalent Cecolin^®^	Bivalent WalrinVax^®^	Quadrivalent Cervavac-4 *
Approval	2006	2009	2014	2019	2022	2022
Manufacturer	Merck & Co., Inc. (Rahway, NJ, USA)	Glaxo Smith Kline plc. (Rixensar, Belgium)	Merck & Co., Inc. (Rahway, NJ, USA)	Xiamen Innovax Co., Ltd. (Xiamen, Fujian, China)	Walvax Co., Ltd. (Kunming, Yunnan, China)	Serum Institute of India (Hadapsar, Pune, India)
VLP HPV Type (protein dose)	HPV-6, 18 (20 µg each) HPV-11, 16 (40 µg each)	HPV-16 (20 µg) HPV-18 (20 µg)	HPV-6 (30 mg) HPV-16 (60 mg) HPV-11, 18 (40 mg each) HPV-31, 33, 45, 52, 58 (20 mg each)	HPV-16 (40 µg) HPV-18 (20 µg)	HPV-16 (40 µg) HPV-18 (20 µg)	HPV-6, 18 (20 µg each) HPV-11, 16 (40 µg each)
Expression system	*Saccharomyces cerevisiae*	*Trichoplusia ni* insect cell line	*Saccharomyces cerevisiae*	*Escherichia coli*	*Pichia pastoris*	*Hansenula* *polymorpha*
Adjuvant	AAHS	AS04	AAHS	Aluminum hydroxide	Aluminum phosphate	Aluminum hydroxide

* Available only in India. AAHS: amorphous aluminum hydroxyphosphate sulfate; AS04: adjuvant System 04, developed by GlaxoSmithKline, composed of a combination of monophosphoryl lipid A and aluminum salt.

**Table 2 viruses-17-01209-t002:** Different types of drugs delivered by VLPs.

VLP Origin	Loaded Material	Application	References
Adenovirus	Bleomycin Paclitaxel mRNA	Tumor therapy	[84,85]
Bacteriophage Qβ	Azithromycin Clarithromycin	Antimicrobial drug	[86]
Bacteriophage MS2	Doxorubicin, Cisplatin, 5-fluorouracil siRNA Ricin toxin A-chain	Tumor therapy	[87]
Cowpea mosaic virus (CPMV)	Doxorubicin	Tumor therapy	[88]
Cucumber mosaic virus (CMV)	Doxorubicin	Tumor therapy	[89]
Filamentous bacteriophages	Chloramphenicol	Antimicrobial drug	[90]
Hepatitis B virus (HBV)	siRNA	Tumor therapy	[91]
Polyomavirus	Methotrexate	Tumor therapy	[92]
Rotavirus	Doxorubicin	Tumor therapy	[93,94]

**Table 3 viruses-17-01209-t003:** VLP types are applied as a vaccine template against distinct types of solid tumors, either associated with or not associated with specific adjuvants or therapies.

VLP Source	Cancer Type	Antigen Target	Association		References
Pre-clinical studies					
AP205	Cervical Breast	HPV RG1 epitope (from L2) HER-2	--		[21,95]
Bacteriophage Qβ	Melanoma	PMEL17, MTC-1, Calpastatin, ZFP518, TRP-2, Caveolin2, Cpsf3l, and Kifl8b	Anti-CD25		[96]
CMV	Melanoma	LCMV-gp33	Microcrystalline tyrosine		[97]
CPMV	Metastatic models	Empty	--		[98]
HBcAg	Hepatocellular carcinoma	MAGE-1, MAGE-3, AFP-1 HBx protein	--		[99,100,101]
Infectious bursal disease virus (IBDV)	Cervical	E7	--		[102]
MS2	Breast Cervical	cystine-glutamate antiporter protein (xCT) L2	-		[103,104,105]
Polyomavirus	Melanoma	OVA (model antigen), TRP2	QuilA-saponin adjuvant or alone		[106]
Rabbit Hemorrhagic Disease Virus (RHDV)	Cervical Colorectal	E6 Topoisomerase IIα and Surviving	Anti-CTLA4 or antiCD25 Unmethylated CpGs		[107]
Simian-Human Immunodeficiency Virus (SHIV)	Pancreatic	hMSLN, mMSLN	--		[108,109]
Simian immunodeficiency virus (SIV)	Pancreatic	Trop2	With gemcitabine or alone		[110]
Clinical trials				Phase	
Bacteriophage Qβ	Melanoma types	--	Pembrolizumab (anti-PD-1)	I	[111,112]
Bacteriophage Qβ	Melanoma Lymphoma	--	Nivolumab (anti-PD1)	II	[113]
Bacteriophage Qβ	Melanoma stage II/IV	Melan A	CpG type A	I/II	[114,115,116]
Bacteriophage Qβ	Melanoma stage II/IV	Melan A	IFA (Montanide) or imiquimod	IIa	[117]
Chimeric HPV16-VLPs	Cervical intraepithelial neoplasia (CIN 2/3)	E7 and 16L1	--	I	[76]

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
