# Peer review of "Recombinant Chimeric Virus-like Particles of Human Papillomavirus Produced by Distinct Cell Lineages: Potential as Prophylactic Nanovaccine and Therapeutic Drug Nanocarriers"

_viruses, 2025, doi:10.3390/v17091209_

Round 1
Reviewer 1 Report
Comments and Suggestions for Authors
Oliveira et al. reviewed L1/2 VLPs as vaccine or drug carriers extensively. This review was generally well-written with good English language quality. I do not have critiques but some general recommendations.
Firstly, the title of this paper is somewhat confusing. Please consider changing the title to something like "Recombinant chimeric human papilomavirus virus-like particles produced in distinct cell lineages: potential as prophylactic nanovaccines and therapeutic drug nanocarriers".
The comparison in efficacy between V1/2 and L1 VLP should be further discussed. For example, L2 provides additional subdominant epitopes that are conserved across HPV types. Has anyone compared the efficacy of L1 VLP and L1/2 VLP in the same study?
The interconnections between sections 3.1/3.2/3.3 and the main topic VLP are unclear. For example, when it comes to section 3.1, I was expecting a topic related to using VLP to deliver DNA rather than DNA-based vaccines for HPV. It was abrupt that the topic was changed.
In section 3.6, further discussion on the advantages of using VLPs as drug carriers particularly for tumor therapy is missing. What made them preferably taken by tumor cells? What effects do VLPs have for different immune cell subsets? Should these immune responses be avoided? How are the generated immune responses against VLPs beneficial for drug delivery? The readers might want to know these.
Reviewer 2 Report
Comments and Suggestions for Authors
Dear authors,
The current review aimed to discuss the potential of recombinant chimeric virus-like particles of HPV produced by distinct cell lineages. Authors suggest that the use of in silico new epitope selection and innovative nanotechnologies can also offer promising therapeutic strategies, encompassing various possibilities for complementary studies to develop potential preventive and therapeutic vaccines with broad-spectrum protection. They suggest the use of VLP as a prophylactic nanovaccine and therapeutic drug nano-carriers. This is a good point for vaccination against HPV and the treatment of cancers caused by this virus.
Authors discussed freely viral structure, pathogenesis, VLPs, Prophylactic and Therapeutic DNA Vaccine against HPV and Virus-Associated Cancers, Some Examples of chi-VLP Applications, Applications of VLPs in Drug Delivery, targeted delivery of therapeutic agents, and many related and unrelated topics.
It is an excellent review, but too long with many headings, subheadings, and sub-sub-subheadings, for example, 2. Overview of basic concepts and 3. 2. Overview of specific concepts. While both discuss the HPV, VLPs. Better to use headings and subheadings only. Also, number 4. Viral Structure and Physical Properties is not needed regarding this topic, and too long.
This review is very important; however, it is too long with too much staffing, and readers may lose interest
Other comments in the attached manuscript
Best regards,

Reviewer 3 Report
Comments and Suggestions for Authors
A few additional paragraphs or section is needed for discussing the potential adverse immune effects of administering larger dose of recombinant chimeric virus-like particles of human papillomavirus and strategies to mitigate them.
